# Effect of Kaolin Clay and ZnO-Nanoparticles on the Radiation Shielding Properties of Epoxy Resin Composites

**DOI:** 10.3390/polym14224801

**Published:** 2022-11-08

**Authors:** Mahmoud I. Abbas, Abdullah H. Alahmadi, Mohamed Elsafi, Sultan A. Alqahtani, Sabina Yasmin, M. I. Sayyed, Mona M. Gouda, Ahmed M. El-Khatib

**Affiliations:** 1Physics Department, Faculty of Science, Alexandria University, Alexandria 21511, Egypt; 2Department of Physics, College of Science, University of Hail, P.O. Box 2440, Hail 81441, Saudi Arabia; 3Department of Physics, Chittagong University of Engineering and Technology, Chattogram 4349, Bangladesh; 4Department of Physics, Faculty of Science, Isra University, Amman 11622, Jordan; 5Department of Nuclear Medicine Research, Institute for Research and Medical Consultations, Imam Abdulrahman bin Faisal University, P.O. Box 1982, Dammam 31441, Saudi Arabia

**Keywords:** epoxy resin, kaolin clay, ZnO-nanoparticles, radiation shielding, point sources

## Abstract

The use of radiation is mandatory in modern life, but the harms of radiation cannot be avoided. To minimize the effect of radiation, protection is required for the safety of the environment and human life. Hence, inventing a better shield than a conventional shielding material is the priority of researchers. Due to this reason, this current research deals with an innovative shielding material named EKZ samples having a composition of (epoxy resin (90–40) wt %-kaolin clay (10–25) wt %-ZnO-nano particles (0–35) wt %). The numerous compositional variations of (epoxy resin, kaolin clay, and ZnO-nano particles on the prepared EKZ samples varied the density of the samples from 1.24 to 1.95 g/cm^3^. The radiation shielding parameter of linear attenuation coefficient (LAC), half value layer (HVL), tenth value layer (TVL), and radiation protection efficiency (RPE) were measured to evaluate the radiation diffusion efficiency of newly made EKZ samples. These radiation shielding parameters were measured with the help of the HPGe detector utilizing the three-point sources (Am-241, Cs-137, and Co-60). The obtained results exposed that the value of linear attenuation coefficient (LAC) and radiation protection efficiency (RPE) was maximum, yet the value of half value layer (HVL), and tenth value layer (TVL), were minimum due to the greater amount of kaolin clay and ZnO-nanoparticles, whereas the amount of epoxy resin was lesser. In addition, it has been clear that as-prepared EKZ samples are suitable for low-dose shielding applications as well as EKZ-35 showed a better shielding ability.

## 1. Introduction

Not only high doses of radiation but also long-time absorption of the low dose radiation has created killing effects on the human body [1]. At present, radiation shielding technology has upheld a vital request for the protection of radiation hazards to both environment and human beings [2]. Epoxy resin is one kind of polymer [3]. Cost-effectiveness and easy production technology make epoxy the most favorable, comprising all other polymers [4]. Because of the low atomic number of elements, polymers independently cannot reduce high-energy incident radiation. The contamination of higher atomic numbered compound polymers provides better radiation shielding ability [5]. Outstanding mechanical strength and stiffness and splendid resistance to heat, immersion, and chemical reactions are the inspiring affectional properties of the polymer [6]. Comprising the conventional shielding material like lead and aluminum epoxy is a low costed product [7]. Though lead is outstanding conventional shielding material, nevertheless, for neutron diffusion, hydrogen-rich polymer materials are supreme [8]. The shielding ability of epoxy resin is approximately similar to concrete. Because of featherweight, polymer aprons or wears are more comfortable than lead aprons; moreover, non-toxicity scientist rushes to improve the shielding ability of polymers [9,10,11].

The US Food and Drug Administration (FDA) has established that usually, ZnO is a harmless material [12]. Due to the advancement of nanomaterials, amplified use of ZnO nanomaterials has been found to shield UV [13]. Nano-ZnO (nZnO) particle has provided boosted execution than macro-ZnO [14]. ZnO has numerous favorable properties, such as being environmentally friendly, ultraviolet-proof, and having a high catalytic activity [15]. Cost-effective and easy production are the most significant characteristics of ZnO nanoparticles comprising other nanoparticles [16]. Nano-ZnO has played a significant role in enriching the compressive and flexural strengths and dry shrinkage reduction of alkali-activated slag (AAS) [17]. Physical and chemical stability, photocatalytic activity, ultraviolet and infrared absorption, antibacterial ability, and non-toxicity are the most exclusive possessions of nano-ZnO [18]. A noteworthy shielding impact has been found for the contamination of nano-sized ZnO between the nano-sized ZnO and micro-sized ZnO in the ceramic [19]. To reduce the radiation effects addition of nano-WO_3_ in epoxy resin showed better performance than micro-WO_3_ [20].

Kaolin is nothing but a summative of the hydrous silicate of alumina presented as Al_2_Si_2_O_5_(OH)_4_ [21]. Because of some adorable features such as nature, cost-effectiveness, high thermo-chemical stability, environment welcoming, corrosion resistance, and outstanding radiation shielding ability, kaolin has become more popular among scientists [22]. In 2021, Echeweozo et al. figured out the gamma radiation defensive ability of baked and unbaked kaolin–granite composite bricks (thickness 3 cm with 0%, 10%, 20%, 30%, 40%, and 50% of granite). Results display that contamination of granite to the kaolin pointedly boosts the shielding capability of kaolin [23]. In the same year, Echeweozo et al. researched the baked and unbaked kaolin powders’ shielding and liquid permeability coefficient. Obtained results indicate that unbaked kaolin is an effective substitute for radiation shielding absorbers having low liquid permeability co-efficiency. In addition, unbaked kaolin has exposed better radiation shielding capability comprising baked kaolin [24]. Recently, Almatari et al. studied the urge of shielding ability of kaolin clay with the contamination of nano- and micro-Bismuth-Oxide. The results revealed that the KC with nano- Bi_2_O_3_ showed healthier radiation defensiveness than KC with micro- Bi_2_O_3_. It is worth noting that a greater amount of Bi_2_O_3_ on the kaolin has shown higher radiation shielding [25]. 

Considering all of these above reasons, a forerunner epoxy has been prepared by adding kaolin and ZnO nanoparticles in the interest of having better radiation shielding efficacy.

## 2. Materials and Methods

### 2.1. Materials

#### 2.1.1. Epoxy Resin

Epoxy Resin is a flexible, transparent liquid, resistant to moisture, heat, and solid at the same time, in addition to that, it lasts more than 20 years. The epoxy resin used in this study is “Conbextra EP10” liquid type, and its properties have been presented in Table 1.

#### 2.1.2. Kaolin Clay

Kaolin is mainly used in the manufacture of porcelain and crockery and in the manufacture of pottery. The used kaolin clay was collected from a quarry in Egypt. To identify its components, first crushed the selected kaolin and sieved them with a 50 μm sieve. It was analyzed by EDX analysis (using an electron microscope with have acceleration voltage of 20.00 kV and magnification × 500) and is also shown in the EDX image in Figure 1. The percentages of compositional oxides in kaolin clay have been presented in Table 2.

#### 2.1.3. ZnO-Nanoparticles

ZnO-Nanoparticles were prepared chemically by Nano Tech Company (a company working on the preparation of nanoparticles in Cairo, Egypt). To prepare ZnO nanoparticles, TEM (JEOL JEM-2100 high-resolution transmission electron microscope type) was performed to produce the average particle size was 30 ± 5 nm. The TEM image of ZnO nanoparticles is displayed in Figure 2.

### 2.2. Samples Preparation 

At first, the epoxy resin was mixed with kaolin clay and then stirred very well. After that, ZnO nanoparticles were added, and 5 wt % of epoxy resin hardener was added to the prepared mixture and stirred to homogenize them. In the second step, homogenized mixture was placed into molds to form a 2 cm diameter of the object. Finally, leave them for 24 h until they dry, and give a pioneer epoxy shield for the radiation shielding application. Six pioneer epoxy samples were made following the compositional ratio, displayed in Table 3 and coded as EKZ-0, EKZ-5, EKZ-10, EKZ-20, EKZ-25, and EKZ-35.

### 2.3. Attenuation Factors Measurements

The attenuation factors of the present developed epoxy samples were experimentally estimated using an HPGe detector with a resolution of 1.92 at 1.333 MeV and relative efficiency of 24%. Point sources of different energy ranges have been used, Am-241 giving a line of 0.060 MeV, Cs-137 giving a line of 0.662 MeV, and Co-60 giving two energy lines of 1.173 and 1.333 MeV [26,27,28]. The absorbed sample is placed between the detector and the source at an appropriate distance (calibrated before the measurements), as shown in Figure 3. The measurement was carried out for a sufficient time in the presence and absence of the sample for all sources with the same conditions to obtain different peaks related to the incident energy photons, whose areas (in the presence (A) and absence ($A_{0}$) of the sample) is determined using the Genie 2000 software. From the calculated areas, the transmission factor (*TF*) can be experimentally evaluated by the following Equation [29,30,31,32]
(1)$$TF\;\left(\%\right)=\frac{I}{I_{0}}\times 100$$
(2)$$TF\;\left(\%\right)=\frac{A}{A_{0}}\times 100$$

The linear attenuation coefficient (*LAC*) represents the interaction potential of a photon inside the prepared marble sample through a certain pathlength (x), and it can be calculated by the following law [33,34,35]:(3)$$LAC=\frac{1}{x}\;Ln\frac{A_{0}}{A}$$

The thicknesses needed to reduce the estimated area to its half and tenth value are called the half (*HVL*) and tenth (*TVL*) value layers, respectively, and are given by [36,37]: (4)$$HVL=\frac{Ln\;2}{LAC}$$
(5)$$TVL=\frac{Ln\;10}{LAC}$$

The Radiation shielding efficiency (*RSE*) of the material represents the material’s ability to absorb the radiation falling on it and depends on several factors, including the density and thickness of the material, and is given by the following relationship [38,39]:(6)$$RSE\;\left(\%\right)=\left[1-\frac{A}{A_{0}}\right]\times 100$$

## 3. Results and Discussion

Figure 4 represents I/I_0_ as a function of the thickness of all the samples at 0.060 MeV. For all the samples, I/I_0_ decreases with increasing thickness, which means that increasing the thickness of the samples reduces the number of photons that can penetrate through the sample, improving the attenuation ability of the sample. For each subfigure, the slope between ln(I/I_0_) and thickness was added, which represents the absolute values of the linear attenuation coefficient (LAC). All the slopes being negative means that the number of photons that penetrate through the sample and the thickness of the sample is inversely related. For example, for EKZ-0, its slope is −0.2712, which means that the LAC for this sample at 0.06 MeV is 0.2712 cm^−1^. When looking at the slopes from EKZ-0 to EKZ-35, in other words, when the ZnO content increases, the slopes also increase, which suggests that the LAC increases with the addition of ZnO nanoparticles.

Appendix A plot I/I_0_ versus thickness at 0.662 MeV, 1.173 MeV, and 1.333 MeV to find the relationship between the LAC values at different energies. Looking at these figures individually, the results are all similar to the trends observed in Figure 4. More specifically, the slopes of all the data points are negative, and they increase with increasing ZnO content. For instance, in Appendix A, EKZ-0’s slope is −0.0766, while EKZ-35’s slope is −0.1147. These results confirm the conclusions made for the samples at 0.060 MeV. When comparing the figures against each other, the slopes for a single sample decrease with increasing energy. For example, EKZ-30’s slopes are equal to −0.1515 at 0.662 MeV, −0.1147 at 1.173 MeV, and −0.1075 at 1.333 MeV. Therefore, these results show that the LAC values of the samples are inversely related to the incoming photon energy.

The LAC, linear attenuation coefficient, and results for all the samples were calculated and graphed in Figure 5. For all six of the investigated samples, LAC and the incoming photon energy have an inverse relationship. More specifically, EKZ-5’s LAC values are equal to 0.379 cm^−1^ at 0.060 MeV, 0.107 cm^−1^ at 0.662 MeV, 0.081 cm^−1^ at 1.173 MeV, and 0.076 cm^−1^ at 1.333 MeV, while EKZ-25’s LAC is equal to 0.939, 0.135, 0.103, and 0.096 cm^−1^ for the same respective energies. This relationship shows that the samples have a better attenuation ability against lower energy photons. The figure also demonstrates that additional kaolin clay and ZnO-nanoparticles lead to an enhancement in the LAC values of the samples. At 0.662 MeV, the LAC values are equal to 0.1006 cm^−1^, 0.1069 cm^−1^, 0.1140 cm^−1^, 0.1249 cm^−1^, 0.1352 cm^−1^, and 0.1515 cm^−1^ for 0, 5, 10, 20, 25, and 35% of ZnO, respectively. This trend has occurred due to the increase of kaolin clay and ZnO-nanoparticles, whereas the amount of epoxy resin has decreased at the same time, resulting in a decrease in the density of the samples, which increases the LAC values as well as the size distribution; when the particles are in the nano size, the particle distribution within the material is better, the absorption ratio increases, and thus the shielding efficiency increases.

Figure 6 shows the HVL, half value layer of the EKZ samples as a function of photon energy. It can be observed that EKZ-0 has the maximum HVL values, while EKZ-35 has the minimum HVL values. For instance, EKZ-0 has HVL values of 2.6 cm at 0.060 MeV and 9.7 cm at 1.333 MeV, while EKZ-35′s HVL values are equal to 0.5 cm and 6.5 cm for the same respective energies. Between EKZ-0 and EKZ-35, the HVL values continuously decrease with increasing ZnO content. More specifically, at 1.173 MeV, the HVL values are equal to 9.1 cm, 8.5 cm, 8 cm, 7.3 cm, 6.8 cm, and 6 cm for EKZ-0 to EKZ-35, respectively. When focusing on one sample at a time, it can be seen that the HVL values increase with energy. EKZ-10′s HVL, for instance, increases from 1.4 cm at energy 0.060 MeV, 6.1 cm at energy 0.662 MeV, 8 cm at energy 1.173 MeV, and 8.5 cm at energy 1.333 MeV. This result indicates that the samples need a greater thickness to attenuate the same quantity of photons at higher energies. Between the sample EKZ-0 and EKZ-35, the HVL values continuously decrease with increasing the kaolin clay and ZnO-nanoparticles. At energy 0.060 MeV, sample EKZ-35 has shown a five times greater value than sample EKZ-0.

Figure 7 displays the HVL at an energy of 0.06 MeV and the density of the EKZ- samples. Sample EKZ-0 has shown the highest value of HVL (9.7 cm), whereas the lowest value is HVL (6.5 cm) at an energy of 0.06 MeV. It has indicated that with the increase of kaolin clay and ZnO-nanoparticles, the value of HVL has decreased. Noteworthily, it has also been seen that according to the enhancement of kaolin clay and ZnO-nanoparticles, the density of the samples has been increased. Maximum density (1.95 g/cm) has been found for sample EKZ-35, while EKZ-5 has given the minimum value of density (1.24 g/cm). It revealed that with the increase of density, the value of HVL has decreased at a specific energy.

The tenth value layer (TVL) of the six EKZ samples has been graphed in Figure 8. against energy. Two trends have been seen in this figure. First, TVL decreases with energy. This is similar to the trend of HVL values; however, all the TVL values for the same samples are higher than their equivalent HVL values at the same energy. For example, at 0.060 MeV, EKZ-0 has a TVL equal to 8.5 cm, EKZ-10 has a TVL equal to 4.6 cm, and EKZ-35 has a TVL equal to 1.8 cm. Noteworthily, ZnO and HVL have shown a positively correlate with each other. EKZ-5′s TVL values are equal to 6.1 cm, 21.6 cm, 28.3 cm, and 30.2 cm at energy 0.060 MeV, 0.662 MeV, 1.173 MeV, and 1.333 MeV, respectively. Thus, increasing the amount of ZnO in the EKZ samples improves the space efficiency of the material.

Figure 9 shows the radiation protection efficiency (RPE) of the EKZ samples as a function of the incoming photon energy. For all four tested energies, the RPE values increase with increasing ZnO content, which is especially evident at 0.060 MeV. At this energy, the RPE values are equal to 41.9% for EKZ-0, 53.1% for EKZ-5, 63.4% for EKZ-10, 77.9% for EKZ-20, 84.7% for EKZ-25, and 92.7% for EKZ-35. The same trends can be observed for the other three energies, although the difference between the values is much smaller. The figure also shows that the RPE of any single sample decreases with increasing energy. For instance, EKZ-5’s RPE decreases from 53% to 14% for the energy 0.060 MeV to 1.333 MeV, respectively, while EKZ-25’s RPE decreases from 85% to 17% at the same respective energies. Therefore, higher energy photons have an easier time penetrating through the samples, although increasing the ZnO content in the samples improves their shielding ability.

## 4. Conclusions

For all the studied samples EKZ-0 to EKZ-35, ln (I/I_0_) decreased with the enhancement of thickness, as the slope between ln(I/I_0_) and thickness represents the absolute values of the linear attenuation coefficient (LAC); hence, it can be stated that the shielding ability of epoxy resin increase with the enhancement of ZnO- nanoparticles and kaolin clay content. The obtained value of LAC displayed that the samples have a better attenuation ability against lower energy photons. The increasing amount of kaolin clay and ZnO-nanoparticles, as well as the decreasing amount of epoxy resin, upsurges the density of the prepared EKZ samples, which surges the value of LAC. It has indicated that with the increase of kaolin clay and ZnO-nanoparticles, the value of HVL has decreased. Noteworthy, it has also been seen that according to the enhancement of kaolin clay and ZnO-nanoparticles, the density of the samples has been increased. The half-value layer (HVL) and tenth value layer (TVL) of the six EKZ samples have also displayed the minimum value for the higher amount of kaolin clay and ZnO-nanoparticles as well as the minimum amount of epoxy resin. The value of RPE (radiation protection efficiency) decreased with increasing energy. Herein, considering all of the measured shielding parameters, sample EKZ-35 exposed the highest shielding ability among the six studied EKZ samples. Therefore, it can be plainly stated that kaolin clay and ZnO-nanoparticles improve the shielding ability of epoxy resin.

## Figures and Tables

**Figure 1 polymers-14-04801-f001:**
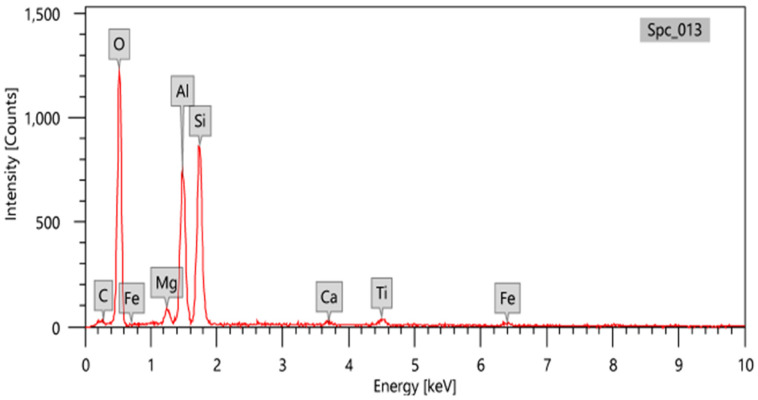
EDX analysis of kaolin clay.

**Figure 2 polymers-14-04801-f002:**
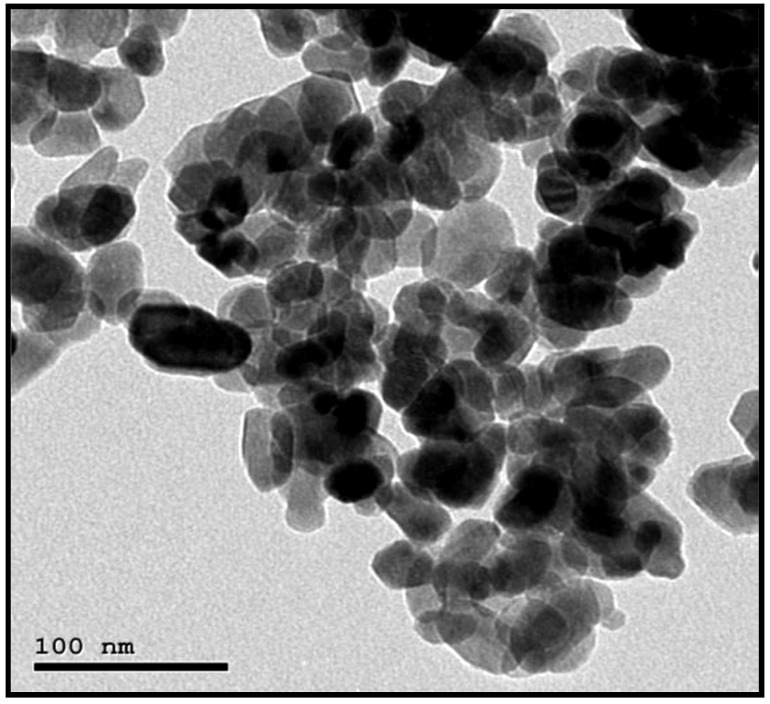
TEM image of ZnO nanoparticles.

**Figure 3 polymers-14-04801-f003:**
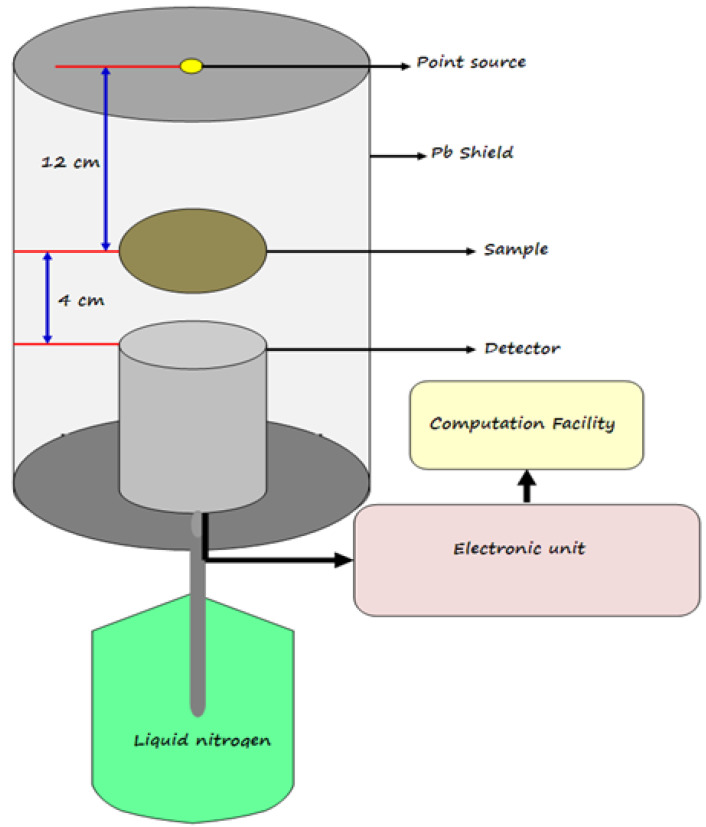
The geometry of the experimental work.

**Figure 4 polymers-14-04801-f004:**
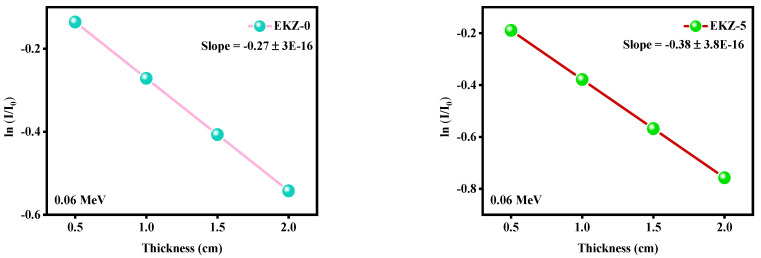
The ln (I/I_0_) versus the thickness for all prepared samples at 0.060 MeV.

**Figure 5 polymers-14-04801-f005:**
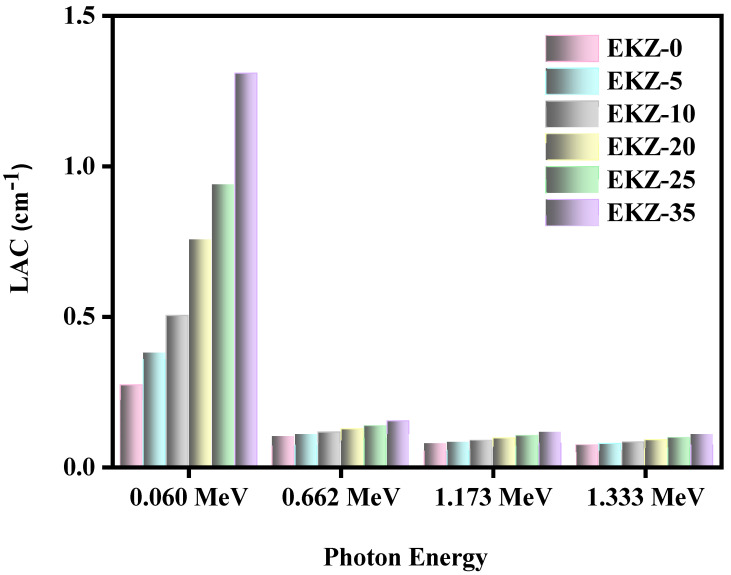
The LAC of EKZ samples as a function of photon energy.

**Figure 6 polymers-14-04801-f006:**
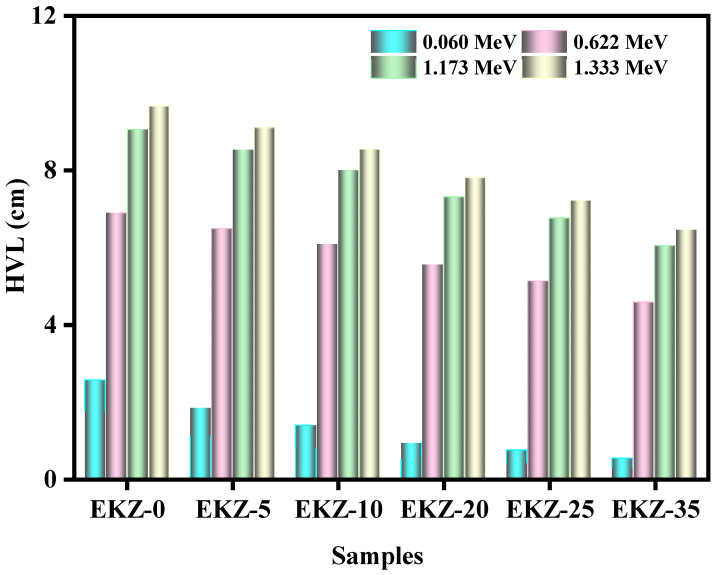
The HVL of EKZ samples as a function of photon energy.

**Figure 7 polymers-14-04801-f007:**
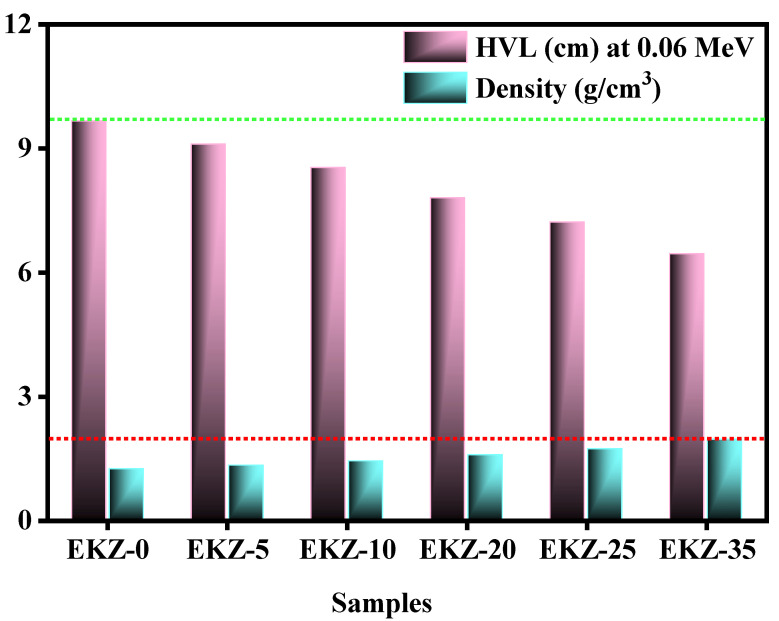
The variation of the HVL and the density of EKZ samples.

**Figure 8 polymers-14-04801-f008:**
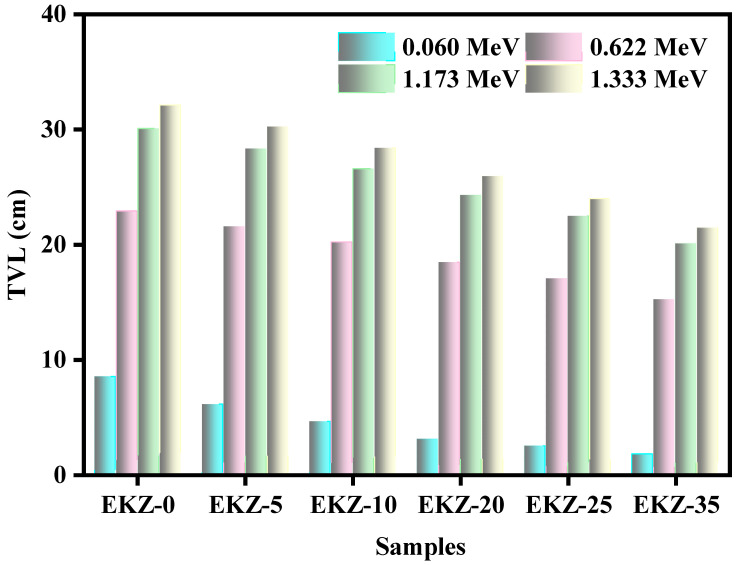
The TVL of EKZ samples as a function of photon energy.

**Figure 9 polymers-14-04801-f009:**
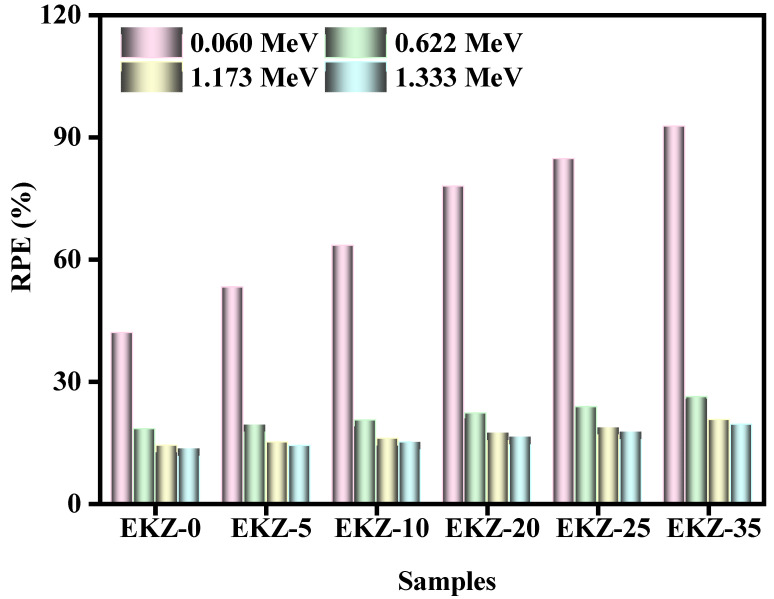
The RPE of EKZ samples as a function of photon energy.

**Table 1 polymers-14-04801-t001:** The properties of used epoxy resin.

Density	1.05 (g/cm^3^)
Compressive Strength	93 (N/mm^2^)
Tensile Strength	26 (N/mm^2^)
Flexural Strength	63 (N/mm^2^)

**Table 2 polymers-14-04801-t002:** The percentages of compositional oxides in kaolin clay.

Oxide	Percentages (%)
MgO	2.99
Al_2_O_3_	35.53
SiO_2_	55.26
CaO	1.24
TiO_2_	2.76
Fe_2_O_3_	2.22

**Table 3 polymers-14-04801-t003:** Chemical compositions of pioneer epoxy samples.

Codes	Compositions (wt, %)			Density (g/cm^3^)
	Epoxy	Kaolin Clay	ZnO-Nanoparticles	
**EKZ-0**	90	10	0	1.237
**EKZ-5**	80	15	5	1.325
**EKZ-10**	70	20	10	1.425
**EKZ-20**	60	20	20	1.577
**EKZ-25**	50	25	25	1.721
**EKZ-35**	40	25	35	1.947

## Data Availability

Not applicable.

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
