# Peer review of "Effect of Kaolin Clay and ZnO-Nanoparticles on the Radiation Shielding Properties of Epoxy Resin Composites"

_polymers, 2022, doi:10.3390/polym14224801_

Round 1

Reviewer 1 Report

Effect of kaoline clay and ZnO nanoparticles on the radiation shielding properties was investigated, and EKZ-35 showed a better shielding ability. The research is imporant for radiation shielding. There are some questions before acceptance. 

(1) The size distribution of ZnO in the EKZ samples. 

(2) The reason of increasing the ZnO content for improving radiation shielding ability. 

(3) References need modification, such as no volume in [7]. 

(4) Table 2, unit is at.%?

Author Response

Effect of kaolin clay and ZnO nanoparticles on the radiation shielding properties was investigated, and EKZ-35 showed a better shielding ability. The research is important for radiation shielding. There are some questions before acceptance. 

Reply: Thank you for your review. All your points were taken and treated in our revision.

(1) The size distribution of ZnO in the EKZ samples. 

Reply: Thank you for your remark. The size distribution was added in the revised version.

(2) The reason of increasing the ZnO content for improving radiation shielding ability. 

Reply: Thank you for your question. The density of ZnO plays an important role for improving the shielding ability as well as the size distribution of ZnO

(3) References need modification, such as no volume in [7]. 

Reply: Thank you for your remark. All references will check by the journal editorial team to make all references match the MDPI reference style before the publication.

(4) Table 2, unit is at. %?

Reply: Thank you for your question. The weight percentage of each component (wt, %)

Reviewer 2 Report

In this article "Effect of kaolin clay and ZnO-nanoparticles on the radiation shielding properties of epoxy resin composites" author mentioned the effect of kaolin clay and ZnO-nanoparticles on the radiation shielding properties of epoxy resin composites. However, this article still has the following issues, and would like to revise them before resubmitting/publishing;

Title should be revised for better explaining the paper content.

Please write authors name and affiliations properly.

A mismatch is also in the number of authors in MDPI system and what mentioned in the manuscript.

Also write title, authors name and affiliation in the supplementary file.

Need to revise the abstract by following a scientific trend.

Revise keywords with some attractive/catchy words. (maximum number should be 5-6).

Some errors regarding the sub/super script, spacing and typo need to consider throughout the manuscript.

If figure 1 is mentioned before Table 1, must arrange accordingly and also check in whole manuscript.

Need to mention about the merits of the chosen material.

 In introduction, mention some other related studies to support your study like Energy and Environment Focus 2 (1), 73-78, Applied Sciences 11 (21), 10451.

Need to improve the language of manuscript and also recheck the full article as there are so many mistakes of typo/formatting/uniformity etc. equation numbers are written irregularly.(3-6)

Need to improve the quality of figures specially font size should be same.

The conclusion must be concised and must be result oriented.

Author Response

In this article "Effect of kaolin clay and ZnO-nanoparticles on the radiation shielding properties of epoxy resin composites" author mentioned the effect of kaolin clay and ZnO-nanoparticles on the radiation shielding properties of epoxy resin composites. However, this article still has the following issues, and would like to revise them before resubmitting/publishing;

Reply: Thank you for your review. All your points were taken and treated in our revision.

Title should be revised for better explaining the paper content.

Reply: Thank you for your remark. The title was revised.

Please write authors name and affiliations properly.

 A mismatch is also in the number of authors in MDPI system and what mentioned in the manuscript.

Also write title, authors name and affiliation in the supplementary file.

Reply: Thank you for your remarks. The authors name and affiliations was added

Need to revise the abstract by following a scientific trend.

Reply: Thank you for your remarks. The abstract was revised

Revise keywords with some attractive/catchy words. (maximum number should be 5-6).

Reply: Thank you for your remarks. The abstract was revised

Some errors regarding the sub/super script, spacing and typo need to consider throughout the manuscript.

Reply: Thank you for your remarks. The whole manuscript was revised to correct the errors, spacing and typos in the manuscript.

If figure 1 is mentioned before Table 1, must arrange accordingly and also check in whole manuscript.

Reply: Thank you for your remark. The whole manuscript was checked.

Need to mention about the merits of the chosen material.

Reply: Thank you for your remark. It was mentioned in section.2

 In introduction, mention some other related studies to support your study like Energy and Environment Focus 2 (1), 73-78, Applied Sciences 11 (21), 10451.

Reply: Thank you for your suggestion. The related studies were mentioned.

Need to improve the language of manuscript and also recheck the full article as there are so many mistakes of typo/formatting/uniformity etc. equation numbers are written irregularly.(3-6)

Reply: Thank you for your remark. The whole article was checked as well by a native English speaker, who has a strong understanding of English grammar. Also the English mistakes and the written were checked carefully through the whole article as well.

Need to improve the quality of figures especially font size should be same.

Reply: Thank you for your remark. The quality of figures was improved.

The conclusion must be concised and must be result oriented.

Reply: Thank you for your remark. The conclusion was revised based on your suggestion.

Reviewer 3 Report

The article presented for review presents very important research because it includes the use of new materials for radiation protection. Such research should be used in the development of standards used for the design of radiation shields. The more so that kaolin clay and ZnO-nanoparticles improve the shielding ability of epoxy resin. However, the article needs a few  adjustments:

1. Please enter the spacing between the text and tables and figures.

2. In the article we have two tables with number 2. This is a problem because there is a reference to table 3 in the text.

3. Please correct the figure 3.

4. In my opinion, figure 4 requires correction and additional description below the drawing.

5. In my opinion, the drawings require standardization (font, signatures).

Therefore, the publication is a valuable source of information to implement new solutions in this field and forms the basis for further research.

Thank you for considering my opinion. I encourage the authors to continue working on improving the manuscript.

Author Response

The article presented for review presents very important research because it includes the use of new materials for radiation protection. Such research should be used in the development of standards used for the design of radiation shields. The more so that kaolin clay and ZnO-nanoparticles improve the shielding ability of epoxy resin. However, the article needs a few adjustments:

Thank you for your positive review. All points were treated based on your suggestions.

  1. Please enter the spacing between the text and tables and figures.

Reply: Thank you for your remark. The spacing between the text and tables and figures was added.

  1. In the article we have two tables with number 2. This is a problem because there is a reference to table 3 in the text.

Reply: Thank you for your remark. It was treated.

  1. Please correct the figure 3.

Reply: Thank you for your remark. It was corrected.

  1. In my opinion, figure 4 requires correction and additional description below the drawing.

Reply: Thank you for your remark. More description was added based on your suggestion.

  1. In my opinion, the drawings require standardization (font, signatures).

Reply: Thank you for your remark. The authors did the best for the paper based on your suggestions.

Therefore, the publication is a valuable source of information to implement new solutions in this field and forms the basis for further research. Thank you for considering my opinion. I encourage the authors to continue working on improving the manuscript.

Thank you for your reviewing our manuscript.

Reviewer 4 Report

The manuscript submitted by Mahmoud I. Abbas et al reported on the “Effect of kaolin clay and ZnO-nanoparticles on the radiation shielding properties of epoxy resin composites” In this work, shielding material called EKZ-samples has been studied and the authors listed several attracting advantages of it. linear attenuation coefficient (LAC), half value layer (HVL), tenth value layer (TVL), and radiation 20 protection efficiency (RPE) were measured to analyze these materials. The paper looks good and detailed experiments were listed. I think minor modification is needed before publication. 

Please double check your author contribution. It seems a template instead of your actual contribution section. Please correct those errors. 

In figure 7, the X axis should be labeled as “samples” 

In all your references, please make them in the same format which “Polymers” requested. Some of your references have the DOI attached but some of them don’t. They are not in the same format. 

Overall, I believe the manuscript and the topic are in an interesting area. I suggest minor revision before publication.

Author Response

The manuscript submitted by Mahmoud I. Abbas et al reported on the “Effect of kaolin clay and ZnO-nanoparticles on the radiation shielding properties of epoxy resin composites” In this work, shielding material called EKZ-samples has been studied and the authors listed several attracting advantages of it. Linear attenuation coefficient (LAC), half value layer (HVL), tenth value layer (TVL), and radiation protection efficiency (RPE) were measured to analyze these materials. The paper looks good and detailed experiments were listed. I think minor modification is needed before publication. 

Thank you for your positive review. All points were treated based on your suggestions.

Please double check your author contribution. It seems a template instead of your actual contribution section. Please correct those errors. 

Reply: Thank you for your remark. It was treated.

In figure 7, the X axis should be labeled as “samples” 

Reply: Thank you for your remark. It was corrected.

In all your references, please make them in the same format which “Polymers” requested. Some of your references have the DOI attached but some of them don’t. They are not in the same format. 

Reply: Thank you for your remark. All references will check by the journal editorial team to make all references match the MDPI reference style before the publication.

Overall, I believe the manuscript and the topic are in an interesting area. I suggest minor revision before publication.

Thank you for your review our manuscript.